

# Effect of premolar extraction and anchorage type for orthodontic space closure on upper airway dimensions and position of hyoid bone in adults: a retrospective cephalometric assessment

Omid Mortezai[1], Zeynab Shalli[1], Maryam Tofangchiha[2], Ahad Alizadeh[3], Francesco Pagnoni[4], Rodolfo Reda[4] and Luca Testarelli[4]

[1] Department of Orthodontics, Dental faculty, Qazvin University of Medical Sciences, Qazvin, Iran
[2] Dental Caries Prevention Research Center, Qazvin University of Medical Sciences, Qazvin, Iran
[3] Medical Microbiology Research Center, Qazvin University of Medical Sciences, Qazvin, Iran
[4] Department of Oral and Maxillo Facial Sciences, Sapienza University of Rome, Rome, Italy

Corresponding authors
Zeynab Shalli, Zey.shalli@gmail.com
Maryam Tofangchiha,
mt_tofangchiha@yahoo.com

## ABSTRACT

**Background.** This study aimed to assess the effect of premolar extraction and anchorage type for orthodontic space closure on upper airway dimensions and position of hyoid bone in adults by cephalometric assessment.

**Methods.** This retrospective study was conducted on 142 cephalograms of patients who underwent orthodontic treatment with premolar extraction in four groups of (I) 40 class I patients with bimaxillary protrusion and maximum anchorage, (II) 40 class I patients with moderate crowding and anchorage, (III) 40 class II patients with maximum anchorage, and (IV) 22 skeletal class III patients with maximum anchorage. The dimensions of the nasopharynx, velopharynx, oropharynx, and hypopharynx, and hyoid bone position were assessed on pre- and postoperative lateral cephalograms using AudaxCeph v6.1.4.3951 software. Data were analyzed by the Chi-square test, paired t-test, and Pearson's correlation test (alpha = 0.05).

**Results.** A significant reduction in oropharyngeal, velopharyngeal, and hypopharyngeal airway dimensions was noted in groups I, III, and IV ($P < 0.001$), which was correlated with the magnitude of retraction of upper and lower incisors ($r = 0.6 - 0.8$). In group II, a significant increase was observed in oropharyngeal and velopharyngeal dimensions ($P < 0.001$). A significant increase in nasopharyngeal dimensions occurred in all groups ($P < 0.001$). Also, in groups I and III, the position of hyoid bone changed downwards and backwards, which was correlated with reduction in airway dimensions ($r = 0.4 - 0.6$).

**Conclusion.** According to the present results, extraction orthodontic treatment affects upper airway dimensions and hyoid bone position. Maximum anchorage decreases airway dimensions while moderate anchorage increases airway dimensions.

# INTRODUCTION

At present, the orthodontic paradigm has shifted towards the soft tissue, and orthodontists believe that soft tissue analysis including assessment of the facial contour, neuromuscular function, tongue, tonsils, and airways is an inseparable part of orthodontic diagnosis and treatment planning (*Han et al., 1991*).

Extraction of permanent teeth as part of orthodontic treatment has always been a challenging topic in clinical orthodontics (*Baumrind et al., 1996*; *Stephens et al., 1993*). The decision regarding tooth extraction as part of orthodontic treatment plan depends on a number of factors such as patient's age, dental arch width, facial profile, magnitude of crowding, and clinician's judgment and preferences (*Tweed, 1944*).

The most common indications of orthodontic tooth extraction include moderate to severe dental crowding, bimaxillary dentoalveolar protrusion, and anteroposterior discrepancies, such as camouflage treatment of class II and class III malocclusions. Depending on the diagnosis and treatment plan, usually 2 or 4 premolars are extracted (*Proffit, 1994*). However, at present, not only esthetics and stability of extraction orthodontic treatments are questionable, but also their effects on temporomandibular joint and upper airway volume are matters of discussion (*Bowman, 1999*; *Bravo et al., 1997*; *Erdinc, Nanda & Dandajena, 2007*; *Martin, Muelas & Viñas, 2006*).

The airways can be divided into two parts of the upper airways (nasal cavity, pharynx, and larynx), and the lower airways (trachea, bronchi, and lungs). According to the Gray's classification, the upper airways include the nasopharynx, oropharynx, and hypopharynx, which are responsible for physiological processes of deglutition, speech, and respiration. Also, a part of oropharynx is referred to as retropalatal area or velopharynx, which is located between the soft palate and the posterior pharyngeal wall (*Palomo et al., 2017*). The most important influential factors in the upper airway morphology include the size of tongue and soft palate, position of lateral pharyngeal wall, and position of maxilla and mandible (*Chen et al., 2012*). Also, patients with different malocclusions have differences in size and position of the maxilla and mandible, and airway soft tissue structures, affecting the upper airway morphology (*Cakarne, Urtane & Skagers, 2003*).

The velopharynx is the narrowest part of the airways, which is the most susceptible to stenosis and obstruction, and can be affected by orthodontic treatment (*Palomo et al., 2017*). One of the most important challenges in orthodontic treatment planning is to answer the question whether tooth extraction with decreased length of dental arch limits the tongue space, and affects the upper airway dimensions or not (*De Souza et al., 2007*; *Kikuchi, 2005*).

Evidence shows that significant dentofacial changes occur following extraction orthodontic treatment, which include changes in skeletal structures, soft tissue profile, and position of incisors, and have the potential to affect the position of the tongue and the pharyngeal space (*Erdinc, Nanda & Dandajena, 2007*; *Chen et al., 2012*). Different therapeutic approaches may have different impacts on the upper airway dimensions. Depending on the share of anterior and posterior segments in space closure, differential space closure can be divided into three groups. In group A mechanics, the extraction space is

mainly closed by retraction of anterior teeth (maximum anchorage). In group B mechanics, the extraction space is closed by equal traction of anterior and posterior segments (moderate anchorage). In group C mechanics, the extraction space is mainly closed by protraction of posterior teeth (minimum anchorage) (*Burstone & Kwangchul, 2015*). Several studies have confirmed that premolar extraction and retraction of incisors with maximum anchorage in orthodontic treatment of bimaxillary dentoalveolar protrusion cases result in a reduction in upper airway dimensions (*AlMaaitah, El Said & Abu Alhaija, 2012*; *Stefanovic et al., 2013*; *Wang et al., 2012*). However, no comprehensive study is available regarding the effect of anchorage type in extraction orthodontic treatment on upper airway dimensions in different types of malocclusion.

The main concern with respect to changes in upper airway dimensions following tooth extraction is related to its adverse effects on sleep quality. Several studies have shown that upper airway stenosis leads to respiratory disorders such as snoring, and obstructive sleep apnea (OSA), which negatively affect the quality of life, and can even be life threatening. Recently, some studies revealed that OSA patients have dentofacial morphological properties related to upper airway stenosis, such as a retruded mandible, increased mandibular plane angle, posterior positioning of the tongue, and long soft palate (*Quan et al., 1999*; *Kerr, 1985*; *Kirjavainen & Kirjavainen, 2007*; *Svaza et al., 2011*; *Wadhawan & Kharbanda, 2013*). Moreover, evidence shows that muscles around the upper airways play a role in position of the hyoid bone, and the hyoid bone is located in a more downward and forward position in OSA patients compared with normal individuals (*Shigeta et al., 2010*; *Tsuda et al., 2011*; *Guttal & Burde, 2013*).

Several imaging modalities may be employed for assessment of airway dimensions, such as fluoroscopy, fiber optic pharyngeoscopy, cephalometry, cone-beam computed tomography (CBCT), and magnetic resonance imaging. Although advanced techniques are increasingly used for this purpose, they are costly and not easily available. Lateral cephalometry is a reliable imaging modality which is commonly requested for assessment of dentoskeletal deformities, and can also be used for assessment of upper airway dimensions (*Palomo et al., 2017*; *Schwab, 1998*).

Considering the confirmed effect of glossopalatal and pharyngeal dimensions on the size and structure of the upper airways, and the possible effect of orthodontic treatment on these dimensions, this study aimed to assess the effect of premolar extraction and anchorage type for orthodontic space closure on upper airway dimensions and position of hyoid bone in adults by cephalometric assessment. Also, such changes were compared in patients with different types of malocclusion.

## MATERIALS & METHODS

This retrospective study was conducted on patients who underwent extraction orthodontic treatment at the Orthodontics Department of School of Dentistry, Qazvin University of Medical Sciences between 2014 and 2020, and successfully accomplished their treatment. The study was approved by the Institutional Review Board of Qazvin University of Medical Sciences (approval number: IR.QUMS.REC.1399.411).
## Sample size

The sample size was calculated to be 142 patients assuming the study power of 80%, alpha = 0.05, standard deviation of 0.428, and mean difference of 1.86 using the sample size formula for paired $t$-test.

## Eligibility criteria

The inclusion criteria were (I) patients who underwent fixed bimaxillary orthodontic treatment with extraction of at least two premolars in one jaw, (II) adult patients with a minimum age of 18 years at the time of treatment onset, and (III) availability of preoperative and postoperative lateral cephalograms of patients with optimal quality.

The exclusion criteria were (I) patients with a history of previous orthodontic treatment, growth modification-functional appliance therapy, or orthognathic surgery, (II) patients with craniofacial anomalies (such as cleft lip and/or palate, or craniofacial syndromes), (III) congenital missing of permanent teeth (except for third molars), (IV) history of permanent tooth extraction (except for premolars), (V) positive medical history of pharyngeal pathologies, adenoidectomy, tonsillar enlargement, mouth breathing, snoring, OSA, or nasal obstruction, and (VI) patients with open bite.

## Data collection

Records of all fixed orthodontic patients treated between 2014 and 2020 were retrieved from the archives of the Orthodontics Department of School of Dentistry, Qazvin University of Medical Sciences after obtaining written informed consent from the patients to use their medical records for research purposes. All lateral cephalograms of patients had been taken under similar conditions with the same X-ray unit (Angell-DF 880) with the exposure settings of 11 mA, 80 kVp, and 11 s time with patient's head in natural head position, relaxed lips, and teeth in occlusion. The magnification of scanner was 0.2 mm, which was taken into account in all measurements. All images were digitized and saved with enlargement factor of 1, and resolution of 875 dpi in TIF format. The digital file of all cephalograms was transferred to AudaxCeph v6.1.4.3951, which supports all different types of cephalometric analyses.

According to the extracted data including type of dentoskeletal discrepancy, pattern of premolar extraction, and anchorage type, the patients were assigned to four groups:

(I)        Forty patients with bimaxillary dentoalveolar protrusion with class I molar and skeletal relationship (mean ANB = 2.3 $\pm$ 0.64 degrees, mean Wits = 1.1 $\pm$ 0.5) treated with maximum anchorage by extraction of 4 maxillary and mandibular first premolars.

(II)       Forty patients with moderate to severe crowding (mean maxillary crowding of 2.3 $\pm$ 5.9 mm, and mean mandibular crowding of 6.3 $\pm$ 2.5 mm), class I molar and skeletal relationship (mean ANB = 2.39 $\pm$ 0.67 degrees, mean Wits = 1 $\pm$ 0.5 mm) treated with moderate anchorage by extraction of 4 maxillary and mandibular first premolars.

(III)      Forty skeletal class 2 division I patients with a mean overjet $\geq$ six mm, and class II molar and skeletal relationship (mean ANB = 4.75 $\pm$ 0.75 degrees,

mean Wits = 5 ± 1.2) treated with maximum anchorage by extraction of maxillary first premolars bilaterally.

(IV)    Twenty-two mild skeletal class III patients with class 3 molar and skeletal relationship (mean ANB = −1.84 ± 0.81 degrees, mean Wits = −1.8 ± 1.01) treated with maximum anchorage by extraction of 2 mandibular first premolars.

Dimensions of the upper airways were measured according to the landmarks and distances mentioned in Table 1 as depicted in Fig. 1 (*Palomo et al., 2017*; *Kirjavainen & Kirjavainen, 2007*; *Lyberg, Krogstad & Djupesland, 1989*; *Joy et al., 2020*; *Bhatia, Jayan & Chopra, 2016*; *Nagmode, Yadav & Jadhav, 2017*). To identify the points corresponding to PTM, U, SP, and Eb on the posterior pharyngeal wall (MPW, UPW, SPW, and LPW), lines parallel to the Gonion-B point horizontal plane were used (*Palomo et al., 2017*).

The position of hyoid bone was also determined according to the cephalometric landmarks and linear measurements as reported in Table 1 and depicted in Fig. 2.

Other dentoskeletal indices were also measured according to the Steiner and Ricketts analyses (*Jacobson & Jacobson, 2006*) (Table 1, Fig. 3).

To measure the magnitude of retraction of incisors and protraction of first molars, a hypothetical vertical line was drawn at point S perpendicular to the Frankfurt plane (Sprep) (*Joy et al., 2020*; *Nagmode, Yadav & Jadhav, 2017*).

To ensure the accuracy of measurements, 10% of cephalograms were randomly selected and the respective variables were manually measured by two examiners (orthodontists). The results were then compared with the results yielded by the software. The mean difference was <1 mm for linear measurements and <1 degree for angular measurements. The standard error for each variable was calculated using paired $t$-test, and the $P$ value was found to be statistically insignificant.

## Statistical analysis

R software was used for statistical analyses (*R Core Team, 2021*). The mixed effects logistic regression was applied to analyze the effect of different variables with the adjusted response variable before the intervention. The Pearson's correlation test was applied to analyze the correlation of variables. The r values between 0.8 to 1 indicated a strong correlation, values between 0.4 to 0.6 indicated a moderate correlation, values between 0.2 and 0.4 indicated a weak correlation, and values between 0 to 0.2 indicated absence of a correlation between two variables. The study groups were compared by the Chi-square test, and paired $t$-test was applied for pairwise comparisons of the variables before and after treatment. $P < 0.05$ was considered statistically significant.

## RESULTS

Lateral cephalograms of 142 patients between 18 to 41 years were evaluated. The maximum treatment duration was 43 months; while, the minimum treatment duration was 25 months. Table 2 presents demographic information of patients.

**Table 1  Cephalometric landmarks and lines used to evaluate changes in hyoid, soft palate and tongue position, upper airway dimensions, and skeletal and dental parameters.**

| Point /line | Definition |
|---|---|
| S | Centre of the sella turcica of the sphenoid bone |
| N | Most anterior point of the frontonasal suture in the midsagittal plane |
| Po | Most superior point of the external auditory meatus |
| Or | Lowest point in the inferior margin of the orbit |
| Point A | Most posterior point in the concavity between anterior nasal spine and the dental alveolus |
| Point B | Most posterior point on the concavity along the anterior surface of the symphysis |
| Go | The most convex point along the inferior border of the ramus |
| M | The most inferior point of the symphysis |
| Rgn | The most posterior point of symphysis |
| H | The most superior and anterior points on the body of the hyoid bone |
| H' | Foot point of perpendicular line from H to Mandibular plan |
| Tt | Tongue tip |
| Th | the superior point of tongue |
| eb | Base of epiglottis |
| U | Tip of soft palate |
| C3 | Antero-inferior limit of the third cervical vertebra |
| ANS | Tip of the anterior nasal spine |
| PNS | Tip of the posterior nasal spine |
| Ptm | Pterygomaxillary fissure, Most inferior point on average right and left outlines of pterygomaxillary fissure. |
| MnPl | Mandibular plane, a line joining M and Go |
| Go–B line | A line joining Go and point B |
| UPW | Upper pharyngeal wall, intersection of a parallel line to Go-B line from ptm with posterior pharyngeal wall. |
| SPW | superior pharyngeal wall, Intersection of a parallel line to Go-B line from sp1 with posterior pharyngeal wall. |
| MPW | middle pharyngeal wall, Intersection of a parallel line to Go-B line from U with posterior pharyngeal wall. |
| LPW | Lower pharyngeal wall, Intersection of a parallel line to Go-B line from eb with posterior pharyngeal wall. |
| U1 | the tip of the upper incisor crown |
| L1 | the tip of the lower incisor crown |
| U6 | the distal point of the upper first molar crown |
| L6 | the distal point of the lower first molar crown |
| Frankfort horizontal plane | Horizontal plane running through porion and orbitale |

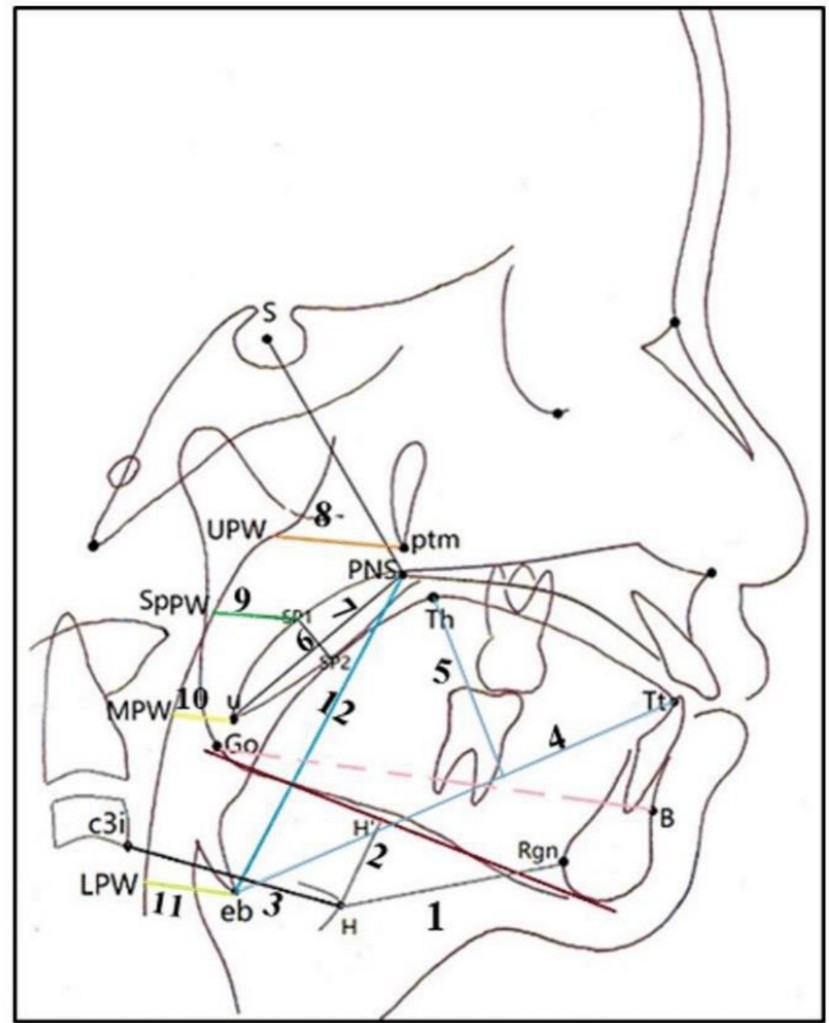

**Figure 1  Lateral cephalometric landmarks, lines, and measurements related to hyoid, tongue and soft palate position and Upper airway dimensions.** 1, H-Rgn, the distance between Rgn and H; 2, H-H', the perpendicular distance from H to the MnPl; 3, HI-C3i, distance between the hyoid bone an C3; 4, TGL, tongue length (eb-Tt); 5, TGH, tongue height (maximum height of the tongue along a perpendicular line of eb-Tt line to tongue dorsum); 6, PNS-U, soft palate length, the distance between PNS and U; 7, sp1-sp2, soft palate thickness (maximum thickness of the soft palate measured on a line perpendicular to PNS-U line): 8, nasopharynx (width of the airway along a parallel line to the Go-B line through ptm) 9, velopharynx, the most constricted airway space (width of the airway behind the soft palate along a parallel line to the Go -Bline); 10, oropharynx (width of the airway along a parallel line to the Go-B line through U); 11, hypopharynx, (width of the airway along a parallel line to the Go-B line through eb), 12, VAL, vertical airway length (the distance between eb and PNS).

## Comparison of upper airway dimensions and position of the tongue, soft palate and hyoid bone after extraction orthodontic treatment

Table 3 presents the changes in upper airway dimensions, and position of hyoid bone, tongue, and soft palate after extraction orthodontic treatment in the four groups.
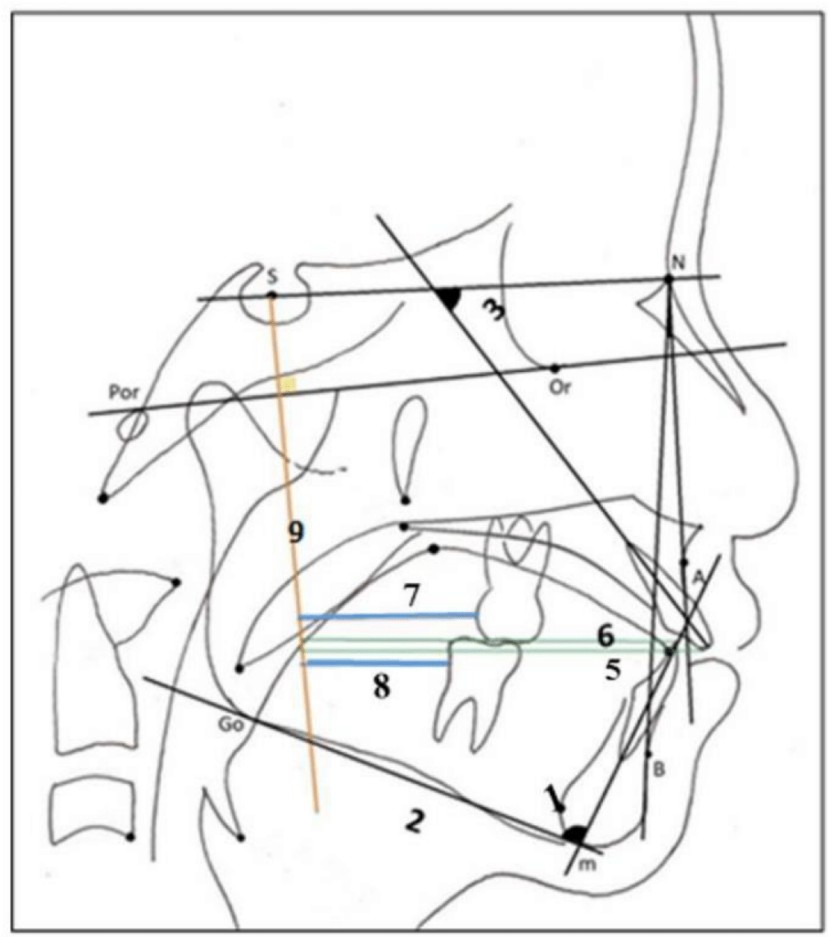

**Figure 2** **Lateral cephalometric landmarks, lines, and measurements related to dentoskeletal parameters.** 1, ANB, Angle between point A and B at nasion; 2, FMA, Angle between mandibular plane and the FH plane; 3, U1/SN, Angle between SN plane and long axis of upper incisors; 4, IMPA. Angle between mandibular plane and long axis of lower incisors; 5, U1- Sprep, Horizontal distance from the perpendicular line through Sella to U1; 6, L1-Sprep, Horizontal distance from the perpendicular line through Sella to L1; 7, U6-Sprep, Horizontal distance from the perpendicular line through Sella to U6; 8, L6-Sprep, Horizontal distance from the perpendicular line through Sella to L6; 9, Sprep, the construction line from S perpendicular to the FH plan.

**Table 2** **Cephalometric landmarks and lines used to evaluate changes in hyoid, soft palate and tongue position, upper airway dimensions, and skeletal and dental parameters.**

| | | Group (1) | Group (2) | Group (3) | Group (4) |
|---|---|---|---|---|---|
| | Frequency(%) | 40 | 40 | 40 | 22 |
| Gender | Female | 25(62.5%) | 24(60%) | 26(65%) | 14(63.6%) |
| | Male | 15(37.5%) | 16(40%) | 14(35%) | 8(36.4%) |
| | Age (year) | 23.3 ± 5.36 | 23.1 ± 4.2 | 24.25 ± 4.05 | 27.64 ± 4.24 |
| Treatment duration (month) | | 32.35 ± 3.76 | 31.3 ± 3.14 | 28.28 ± 2.47 | 31.17 ± 3.97 |

Mortezai et al. (2023), *PeerJ*, DOI 10.7717/peerj.15960

**Table 3  Intra-group and inter-group comparison of pre- and post-treatment mean of upper airway dimensions, hyoid, tongue and soft palate position in each study group.**

| Parameters | Group 1 (n = 40) | | | | Group 2 (n = 40) | | | | Group 3 (n = 40) | | | | Group 4 (n = 22) | | | |
|---|---|---|---|---|---|---|---|---|---|---|---|---|---|---|---|---|
| | Pre | Post | Change (95%CL) | p-value | Pre | Post | Change (95%CL) | p-value | Pre | Post | Change (95%CL) | p-value | Pre | Post | Change (95%CL) | p-value |
| UPW-ptm | 23.72 ± 1.76 | 23.91 ± 1.77 | 0.19 (0.13, 0.26) | <0.001 | 22.34 ± 2.05 | 22.47 ± 2.03 | 0.13 (0.09, 0.16) | <0.001 | 18.59 ± 1.9 | 18.69 ± 1.9 | 0.1 (0.08, 0.12) | <0.001 | 24.41 ± 1.8 | 24.42 ± 1.8 | 0.01 (0, 0.02) | 0.1 |
| SPW-sp1 | 10.04 ± 0.62 | 9.88 ± 0.63 | −0.15 (−0.18, −0.12) | <0.001 | 10.18 ± 0.7 | 11 ± 0.63 | 0.82 (0.74, 0.91) | <0.001 | 8.63 ± 0.46 | 7.68 ± 0.43 | −0.95 (−1.05, −0.86) | <0.001 | 10.95 ± 0.75 | 11.12 ± 0.71 | 0.17 (0.13, 0.21) | 0.054 |
| MPW-U | 8.6 ± 0.59 | 7.33 ± 0.47 | −1.27 (−1.41, −1.14) | <0.001 | 8.84 ± 0.57 | 9.82 ± 0.41 | 0.98 (0.88, 1.08) | <0.001 | 7.98 ± 0.51 | 6.73 ± 0.48 | −1.25 (−1.36, −1.14) | <0.001 | 10.14 ± 0.69 | 10.2 ± 0.67 | 0.06 (0, 0.13) | 0.075 |
| LPW-eb | 9.98 ± 0.67 | 8.53 ± 0.78 | −1.45 (−1.6, −1.31) | <0.001 | 10.11 ± 0.48 | 10.27 ± 0.48 | 0.15 (0.13, 0.18) | <0.001 | 8.06 ± 0.38 | 7.91 ± 0.38 | −0.16 (−0.18, −0.13) | <0.001 | 11.2 ± 0.58 | 11.07 ± 0.6 | −0.13 (−0.15, −0.11) | <0.001 |
| VAL | 56.62 ± 2.01 | 56.79 ± 2.01 | 0.17 (0.14, 0.2) | 0.1 | 57.1 ± 1.88 | 58.9 ± 1.92 | 1.83 (1.65, 2) | <0.001 | 50.91 ± 1.21 | 51.09 ± 1.2 | 0.18 (0.13, 0.23) | 0.15 | 62.85 ± 1.2 | 62.98 ± 1.21 | 0.14 (0.09, 0.18) | 0.23 |
| H-H' | 11.72 ± 0.92 | 11.52 ± 0.95 | −0.2 (−0.24, −0.16) | <0.001 | 12.19 ± 0.77 | 12.5 ± 0.78 | 0.31 (0.26, 0.36) | <0.001 | 10.17 ± 0.45 | 10.44 ± 0.43 | 0.28 (0.24, 0.31) | <0.001 | 13.45 ± 0.64 | 13.25 ± 0.68 | −0.2 (−0.25, −0.14) | <0.001 |
| H-C3i | 30.45 ± 1.24 | 29.71 ± 1.19 | −0.74 (−0.87, −0.61) | <0.001 | 30.67 ± 1.25 | 30.58 ± 1.32 | −0.09 (−0.34, −0.01 ) | <0.001 | 27.48 ± 1.09 | 27.35 ± 1.11 | −0.12 (−0.15, −0.1) | <0.001 | 33.68 ± 1.77 | 34.81 ± 1.85 | 1.13 (0.06, 1.2) | <0.001 |
| H-Rgn | 38.19 ± 1.44 | 38.34 ± 1.45 | 0.16 (0.13, 0.18) | <0.001 | 38.2 ± 1.32 | 38.39 ± 1.32 | 0.19 (0.16, 0.22) | <0.001 | 30.42 ± 1.9 | 30.25 ± 1.92 | −0.17 (−0.2, −0.14) | <0.001 | 40.13 ± 1.007 | 39.98 ± 0.99 | −0.14 (−0.18, −0.11) | <0.001 |
| TGL | 60.37 ± 6.04 | 57.29 ± 5.77 | −3.08 (−3.39, −2.77) | <0.001 | 64.38 ± 6.11 | 64.18 ± 6.16 | −0.2 (−0.24, −0.16) | <0.001 | 59.52 ± 5.57 | 59.32 ± 5.59 | −0.19 (−0.24, −0.15) | <0.001 | 69.63 ± 4.28 | 67.14 ± 4.42 | −2.49 (−2.94, −2.03) | <0.001 |
| TGH | 23.77 ± 3.46 | 23.69 ± 3.48 | −3.08 (−3.39, −2.77) | <0.001 | 22.9 ± 2.21 | 22.8 ± 2.23 | −0.1 (−0.13, −0.08) | <0.001 | 23.7 ± 1.73 | 23.62 ± 1.74 | −0.08 (−0.1, −0.05) | <0.001 | 26.11 ± 2.7 | 26.24 ± 2.69 | 0.12 (0.08, 0.17) | <0.001 |
| Sp1-sp2 | 6.9 ± 1.51 | 7.02 ± 1.52 | 0.13 (0.1, 0.15) | <0.001 | 6.91 ± 0.9 | 7.05 ± 0.93 | 0.14 (0.11, 0.16) | <0.001 | 8.39 ± 1.21 | 8.52 ± 1.24 | 0.13 (0.11, 0.16) | <0.001 | 7.47 ± 1.21 | 7.6 ± 1.23 | 0.13 (0.1, 0.16) | <0.001 |
| PNS-U | 30.28 ± 3.77 | 30.19 ± 3.77 | −0.09 (−0.1, −0.07) | <0.001 | 29.49 ± 4.01 | 29.39 ± 2.02 | −0.1 (−0.11, −0.08) | <0.001 | 28.21 ± 4.45 | 28.08 ± 4.43 | −0.13 (−0.16, −0.09) | <0.001 | 33.36 ± 3.42 | 32.96 ± 4.18 | −0.39 (−0.96, 0.17) | 0.16 |

**Notes.**

Values are Mean ± Standard Deviation. *P*-values by paired *t*-test after confirming the underlying normality assumption of difference in each measurement. *P* value <0.05 is statistically significant.
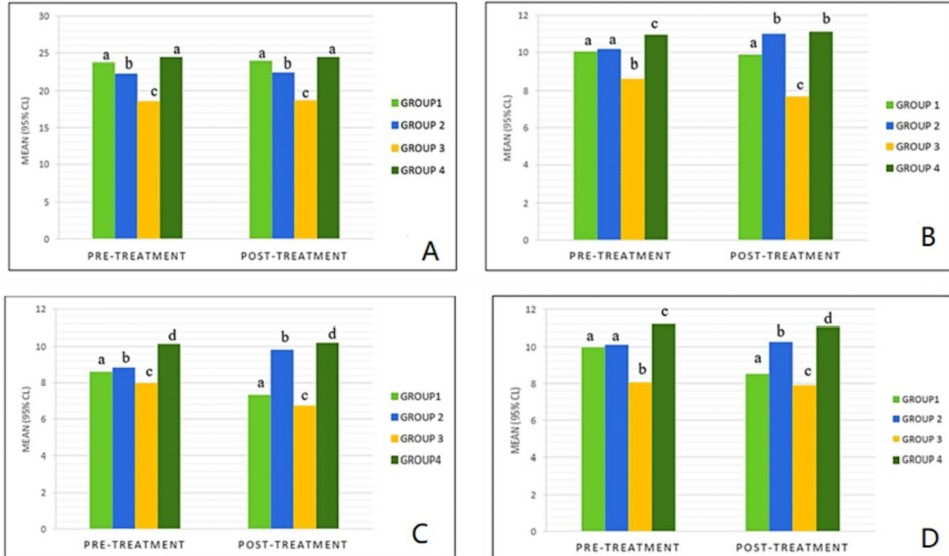

**Figure 3 Comparison of airway dimensions after extraction orthodontic treatment compared with baseline in the study groups.** (A) Mean width of nasopharynx before and after treatment; (B) mean width of velopharynx before and after treatment, (C) mean width of oropharynx before and after treatment, (D) mean width of hypopharynx before and after treatment. Similar letters indicate presence of a significant difference ($P < 0.05$).

In group 1, the mean width of nasopharynx significantly increased while the mean width of oropharynx, velopharynx, and hypopharynx significantly decreased after extraction orthodontic treatment ($P < 0.05$). The changes in position of the tongue, soft palate, and hyoid bone were statistically significant after treatment ($P < 0.05$); however, the change in vertical airway length was not significant ($P > 0.05$).

In group 2, the mean width of nasopharynx, oropharynx, velopharynx, and hypopharynx and vertical airway length significantly increased. Also, the changes in position of the tongue, soft palate, and hyoid bone were statistically significant after treatment ($P < 0.05$).

In group 3, the mean width of nasopharynx significantly increased, while the mean width of velopharynx, oropharynx, and hypopharynx significantly decreased. Also, the changes in position of the tongue and soft palate were significant after treatment ($P < 0.05$).

In group 4, the changes in width of hypopharynx, tongue position, soft palate thickness, and hyoid bone position were significant after treatment ($P < 0.05$). However, the changes in width of nasopharynx, velopharynx, and oropharynx were not significant ($P > 0.05$).

Age at the time of treatment onset and duration of treatment had the same effect in the four groups on upper airway dimensions and hyoid bone position after extraction orthodontic treatment ($P > 0.05$). However, the effect of gender on upper airway dimensions was significant before and after extraction orthodontic treatment in all four groups ($P < 0.001$), such that the mean oropharyngeal and hypopharyngeal airway dimensions were larger in males than females both before and after orthodontic treatment

($P < 0.05$); whereas, the largest mean dimensions of nasopharynx were recorded in females before, and in males after treatment.

Moreober, a moderate correlation existed between the change in hyoid bone position and reduction of upper airway dimensions in groups I, III and IV ($r = 0.55$, $P < 0.05$). Figure 3 compares upper airway dimensions before and after extraction orthodontic treatment in the four groups.

### Comparison of dentoskeletal indices after extraction orthodontic treatment

Table 4 presents the changes in dentoskeletal indices after extraction orthodontic treatment in the four groups. The Pearson's correlation coefficient revealed a positive strong correlation between the retraction of maxillary and mandibular incisors and reduction in oropharyngeal, velopharyngeal, and hypopharyngeal airway dimensions in groups I, III and IV ($P < 0.05$). Also, a significant moderate correlation was found between protraction of maxillary and mandibular molar teeth and increased oropharyngeal, and velopharyngeal airway dimensions in group II ($r = 0.4-0.6$).

## DISCUSSION

This retrospective study assessed the effect of premolar extraction and anchorage type for orthodontic space closure on upper airway dimensions and position of hyoid bone in adults by cephalometric assessment. Also, such changes were compared in patients with different types of malocclusion.

### Changes in upper airway dimensions

In group I (patients with bimaxillary protrusion and maximum anchorage), a reduction was observed in oropharyngeal, velopharyngeal, and hypopharyngeal airway dimensions after orthodontic treatment, which may be attributed to significant retraction of incisors, and narrowing of the tongue space. The tongue is attached to the hyoid bone through several muscular and connective tissue attachments, and thus, posterior movement of the tongue results in airway narrowing (*Germec-Cakan, Taner & Akan, 2011*). *Hwang et al. (2010)* and *Enacar et al. (1994)* demonstrated that upper airway narrowing following mandibular setback surgery occurred as the result of posterior displacement of the tongue. Consistent with the present results, *Bhatia, Jayan & Chopra (2016)* and *Wang et al. (2012)* reported a reduction in sagittal dimensions of the velopharyngeal and hypopharyngeal airways after extraction treatment in patients with bimaxillary protrusion. However, they did not observe any change in nasopharyngeal dimensions. *Nagmode, Yadav & Jadhav (2017)* reported a significant reduction in velopharyngeal and hypopharyngeal dimensions, and an increase in nasopharyngeal dimensions after extraction orthodontic treatment in patients with bimaxillary protrusion. This increase was attributed to adenoid retrusion, which was in line with the present findings, and can have a positive impact on the airways, which is important in patients with respiratory problems. Similarly, *Germec-Cakan, Taner & Akan (2011)* reported a reduction by $3.3 \pm 3.8$ mm in oropharyngeal and hypopharyngeal airway dimensions in patients with bimaxillary protrusion and maximum anchorage, which was
**Table 4  Intra-group and inter-group comparison of pre- and post-treatment mean of dentoskeletal parameters in each study group.**

| Dentoskeletal parameters | Group 1 (n = 40) | | | Group 2 (n = 40) | | | Group 3 (n = 40) | | | Group 4 (n = 22) | | |
|---|---|---|---|---|---|---|---|---|---|---|---|---|
| | Pre | Post | *p*-value | Pre | Post | *p*-value | Pre | Post | *p*-value | Pre | Post | *p*-value |
| ANB | 2.3 ± 0.64 | 2.16 ± 0.63 | <0.001 | 2.39 ± 0.67 | 2.24 ± 0.65 | <0.001 | 4.75 ± 0.75 | 3.23 ± 0.76 | <0.001 | −1.84 ± 0.81 | −1.27 ± 0.94 | <0.001 |
| FMA | 26.38 ± 3.37 | 26.51 ± 3.29 | 0.009 | 25.41 ± 3.03 | 25.16 ± 3.05 | <0.001 | 23.34 ± 2.1 | 23.45 ± 2.06 | <0.001 | 28.37 ± 2.23 | 28.46 ± 2.26 | <0.001 |
| IMPA | 101.68 ± 2.61 | 91.9 ± 1.4 | <0.001 | 96.31 ± 1.35 | 89.88 ± 1.26 | <0.001 | 113.6 ± 2.33 | 103.3 ± 0.96 | <0.001 | 105.9 ± 2.77 | 106.49 ± 3.6 | 0.083 |
| U1/SN | 115.62 ± 3.53 | 103.57 ± 1.35 | <0.001 | 106.52 ± 1.05 | 100.75 ± 0.83 | <0.001 | 95.68 ± 2.83 | 95.51 ± 2.84 | 0.072 | 99.18 ± 1.09 | 91.79 ± 1.37 | <0.001 |
| U1-Sprep | 67.8 ± 2.13 | 61.55 ± 2.15 | <0.001 | 65.72 ± 3.04 | 62.92 ± 3.01 | <0.001 | 75.6 ± 3.25 | 69.41 ± 3.19 | <0.001 | 74.8 ± 3.25 | 74.91 ± 3.33 | 0.091 |
| L1-Sprep | 66.46 ± 3.8 | 59.58 ± 3.71 | <0.001 | 61.52 ± 4.04 | 56.85 ± 4.24 | <0.001 | 35.21 ± 2.84 | 35.56 ± 2.91 | 0.13 | 22.03 ± 1.83 | 22.03 ± 1.83 | 0.76 |
| U6-Sprep | 32.71 ± 3.46 | 33.03 ± 3.41 | 0.603 | 26.67 ± 3.75 | 29.71 ± 3.62 | <0.001 | 68.47 ± 4.4 | 68.5 ± 4.39 | 0.61 | 75.27 ± 4.34 | 70.13 ± 4.15 | <0.001 |
| L6-Sprep | 33.74 ± 3.5 | 34.14 ± 3.41 | 0.06 | 27.57 ± 3.63 | 30.65 ± 3.56 | <0.001 | 36.97 ± 2.99 | 36.97 ± 2.99 | 0.85 | 24.7 ± 1.96 | 25.5 ± 1.97 | 0.062 |

**Notes.**

Values are Mean ± Standard Deviation. *P*-values by paired *t*-test after confirming the underlying normality assumption of difference in each measurement. *P* value <0.05 is statistically significant.

larger than the value found in the present study (1.45 mm). The age range of patients was lower in the study by *Germec-Cakan, Taner & Akan (2011)*; nonetheless, they reported that growth and development had no or insignificant effect on upper airway sagittal dimensions. In contrast to the present findings, *Joy et al. (2020)* and *Valiathan et al. (2010)* detected no significant change in pharyngeal airway dimensions after extraction orthodontic treatment. It should be noted that they used CBCT for airway assessments. In the present study, an increase occurred in nasopharyngeal, velopharyngeal, and oropharyngeal airway dimensions in group II (moderate anchorage), which was in agreement with the results of *Germec-Cakan, Taner & Akan (2011)* who reported 1.5 mm increase in nasopharyngeal, velopharyngeal, and oropharyngeal dimensions after extraction orthodontic treatment in patients with moderate crowding and minimum anchorage; the value reported in their study was slightly higher than the value found in the present study (0.98). This difference can be attributed to different anchorage types.

In group III in the present study (class 2 patients with maximum anchorage), a significant reduction occurred in upper airway dimensions; the reduction in velopharynx dimensions in this group was greater than that in other groups. *Hang & Gelb (2017)* indicated a significant reduction in airway dimensions following extraction orthodontic treatment in class 2 patients. In line with the available literature *El & Palomo (2013)*, the present results showed that skeletal class 2 patients had smaller airway dimensions than other groups even before treatment. Thus, the decision regarding extraction orthodontic treatment plan in such patients requires utmost attention.

Moreover, the present results revealed a significant difference among different malocclusion types regarding the role of extraction orthodontic treatment in reduction of airway dimensions. However, unlike the present study, *Alkawari et al. (2018)* demonstrated an increase in nasopharyngeal and a reduction in hypopharyngeal and velopharyngeal airway dimensions in all class 2 and class 3 patients, and found no significant difference between the two groups. This controversy is probably due to the fact that *Alkawari et al. (2018)* evaluated growing patients, and did not assess the type of anchorage.

The present study revealed significant changes in position of the tongue and soft palate after extraction orthodontic treatment; similar changes were observed by *Aldosari et al. (2020)* but were not statistically significant. This difference can be explained by the larger sample size in the present study.

## Correlation of changes in airway dimensions with dentoskeletal parameters

In group II, a significant correlation was noted between protraction of maxillary and mandibular molars and increased upper airway dimensions. A possible explanation for this finding may be the increased posterior tongue space after mesial movement of molar teeth, which increases the velopharyngeal and oropharyngeal dimensions. Also, a reduction in FMA angle was noted, which can be translated to counterclockwise rotation of the mandibular plane, and was significantly correlated with increased velopharyngeal and oropharyngeal airway dimensions. The reason may be that this rotation results in forward

movement of the mandible, and mesial displacement of molars increases the upper airway dimensions (*Germec-Cakan, Taner & Akan, 2011*).

In groups I and III, a strong correlation existed between retraction of upper and lower incisors and reduction of upper airway dimensions. It may be concluded that significant retraction of anterior teeth after premolar extraction would result in backward movement of the tongue and subsequent compression of the soft palate and narrowing of the upper airways (*Shigeta et al., 2010*). Consistent with the present results, *Wang et al. (2012)* reported that the reduction in velopharyngeal and oropharyngeal airway dimensions was correlated with the magnitude of retraction of incisors. *Chen et al. (2012)* found a correlation between the reduction in minimum cross-sectional area of the airways and upper incisor retraction. Both the abovementioned studies used CBCT for their measurements.

## Changes in hyoid bone position

Another possible explanation for reduction in upper airway dimensions following extraction orthodontic treatment is posterior movement of the hyoid bone (*Ng, Song & Yap, 2019*). Evidence shows that muscles around the upper airways affect the position of hyoid bone, and the hyoid bone has a more inferior position in OSA patients than normal individuals (*Shigeta et al., 2010*; *Tsuda et al., 2011*; *Guttal & Burde, 2013*). The present results revealed backward and downward displacement of hyoid bone after extraction orthodontic treatment in patients in groups I and III, and this positional change was statistically significant. Also, the present results revealed a significant correlation between changed position of the hyoid bone and reduction in upper airway dimensions. In line with the present results, *Chen et al. (2012)* showed a significant correlation between the magnitude of backward displacement of hyoid bone in anteroposterior direction and reduction in upper airway dimensions in patients with bimaxillary protrusion with maximum anchorage. However, *Wang et al. (2012)* reported that the change in hyoid bone position was not significant post-treatment, and downward movement of the hyoid bone is an adaptive reaction to prevent the tongue invading the pharyngeal airway, and has no significant role in reduction of airway dimensions. To assess the change in hyoid bone position, using a stable horizontal or vertical reference plane or an independent landmark which is not influenced by orthodontic treatment such as the H-C3i is more conservative (*Chen et al., 2012*; *Patel et al., 2017*; *Keum et al., 2017*), which experienced a significant change after treatment in the present study.

Use of cephalograms instead of CBCT scans was the main limitation of this study. Obviously, CBCT can provide more accurate information about airway dimensions and changes caused by orthodontic treatment. However, due to its high cost, it is not routinely requested for orthodontic patients in Iran. Nonetheless, it should be noted that a strong correlation has been reported between pharyngeal airway dimensions measured on lateral cephalograms and on computed tomography scans (*Riley, Powell & Guilleminault, 1989*; *Perrotti et al., 2021*). Also, the present study only assessed the morphological changes of the airways in anteroposterior dimension, and the correlation of respiratory function with premolar extraction and incisor retraction was not assessed. Since

respiratory function has a more important correlation with the severity of OSA rather than morphological changes (*Wootton et al., 2014*), future studies can address this correlation by using polysomnography.

## CONCLUSIONS

Within the limitations of this study, the results showed that extraction orthodontic treatment and high retraction of maxillary and mandibular incisors with maximum anchorage in adults with bimaxillary protrusion can lead to backward and downward movement of the tongue, downward and backward movement of the hyoid bone, significant reduction of upper airway dimensions in the oropharynx, velopharynx, and hypopharynx, and significant increase in nasopharyngeal dimensions. The effect of extraction orthodontic treatment on upper airway dimensions was significantly different in different malocclusion types. Maximum reduction was noted in velopharynx after extraction orthodontic treatment of skeletal class 2 patients. Type of anchorage for space closure affected upper airway dimensions postoperatively. Extraction orthodontic treatment with moderate anchorage and mesial movement of molars in adults with class I malocclusion and crowding probably increases the posterior tongue space and subsequently the upper airway dimensions.

## ACKNOWLEDGEMENTS

The authors would like to thank Dr. Mojdeh Kalantar Motamedi for her assistance in writing this paper.

### Funding

The authors received no funding for this work.

### Competing Interests

The authors declare there are no competing interests.

### Author Contributions

- Omid Mortezai conceived and designed the experiments, prepared figures and/or tables, and approved the final draft.
- Zeynab Shalli performed the experiments, authored or reviewed drafts of the article, and approved the final draft.
- Maryam Tofangchiha conceived and designed the experiments, prepared figures and/or tables, and approved the final draft.
- Ahad Alizadeh analyzed the data, prepared figures and/or tables, and approved the final draft.
- Francesco Pagnoni performed the experiments, authored or reviewed drafts of the article, and approved the final draft.

- Rodolfo Reda analyzed the data, authored or reviewed drafts of the article, and approved the final draft.
- Luca Testarelli analyzed the data, authored or reviewed drafts of the article, and approved the final draft.

## Human Ethics

The following information was supplied relating to ethical approvals (*i.e.*, approving body and any reference numbers):

Institutional Review Board of Qazvin University of Medical Sciences

## Ethics

The following information was supplied relating to ethical approvals (*i.e.*, approving body and any reference numbers):

Institutional Review Board of Qazvin University of Medical Sciences

## Data Availability

The raw data and code that support the findings of this study are available in the Supplemental Files.

## Supplemental Information

Supplemental information for this article can be found online at http://dx.doi.org/10.7717/peerj.15960#supplemental-information.

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
