# Peer review of "Effect of premolar extraction and anchorage type for orthodontic space closure on upper airway dimensions and position of hyoid bone in adults: a retrospective cephalometric assessment"

_PeerJ, doi:10.7717/peerj.15960_

## Round 0.1 · original submission · Minor Revisions

The research is original and in line with the journal's goals. A well-defined, relevant, and meaningful research question. As stated in the introduction, the study enriches the literature . the Methods section is well described and has enough detail and information to be replicated.

·

Basic reporting

The English language is clear and the text is easy to read.

No comment.

Experimental design

Contemporary cephalometric analysis has been used. The number of patients is equally decided on 3 groups, as well as there is a 4th group of Class IiI patients which is a good sign for the researchers since Class III patients are very rear and difficult to collect for a research. So, this fact makes the statistic results more valuable and reliable.

No comment

Validity of the findings

No comment

Additional comments

Very interesting and easy to read article on a contemporary topic such as reduction of the airways due to tooth extraction and low tongue position.
Congratulations on your research and good statistics.

Reviewer 2 ·

Basic reporting

The English used in the main text is clear and professional. Although about 30% of the references are older than 20 years, the literature references give sufficient background/context information.

The structure of figures and tables is very clear and they concisely and comprehensively express the results.
However, you should check the correspondence between the number of tables mentioned in the text and the contents of the tables and captions. for example in lines 177, 181, 183 you refer to the contents of table 1 by describing it different from the caption, also in lines 203-204 you say that "table 2 presents demographic information of patients" but it should be table 1 and so on.

Experimental design

The Research is original and in line with the objectives of the journal, Well-defined, relevant and meaningful Research question. As stated in the introduction, the study enriches the literature by adding a comprehensive study on the effect of anchorage type in extractive orthodontic treatment on upper airway dimensions in different types of malocclusion. the Methods section is well described and presents sufficient detail and information to be replicated.

Validity of the findings

All data and results are robust and clear. the statistical analysis used is appropriate
The conclusions answer the original research question.

---

## Round 0.2 · accepted · Accept

The research is original and in line with the objectives of the journal. A well-defined, relevant and meaningful research question. As stated in the introduction, the study enriches the literature. the Methods section is well described and contains sufficient detail and information to be replicated. With the revisions made, the article is ready for publication.

·

Basic reporting

The article is written in an excellent professional English and it was very easy to be read throughout. THe article contains sufficient figures and drawings that easy to understand.

Experimental design

Methods are described sufficiently and are illustrated with drawings.

Validity of the findings

Conclusions are well stated and the results are very interesting for the audience.

Additional comments

Up-to-date research with great importance for the quality of life of the orthodontic patients regarding the cases with extraction of teeth and the space for the tongue which corresponds with the sleeping apnea and snooring.